# Influence of the Method of Preparation of the Pd-Bi/Al$_2$O$_3$ Catalyst on Catalytic Properties in the Reaction of Liquid-Phase Oxidation of Glucose into Gluconic Acid

**Mariya P. Sandu** [1,2,*], **Vladimir S. Sidelnikov** [1], **Andrej A. Geraskin** [3], **Aleksandr V. Chernyavskii** [3] **and Irina A. Kurzina** [1]

[1] Department of Physical and Colloid Chemistry, National Research Tomsk State University; 36 Lenin Ave., 634050 Tomsk, Russia; Vladimir.svibla.sidelnikov@gmail.com (V.S.S.); kurzina99@mail.ru (I.A.K.)
[2] Department of Chemistry, Siberian State Medical University; 2 Moskovsky tract, 634050 Tomsk, Russia
[3] Nanocenter, MIREA, Russian Technological University; 78 Vernadsky Ave., 119454 Moscow, Russia; geraskin_a@mirea.ru (A.A.G.); nc@mirea.ru (A.V.C.)
[*] Correspondence: mpsandu94@gmail.com; Tel.: +7-953-915-6971

**Abstract:** Gluconic acid and its derivatives are extensively used in pharmaceutical, food, textile, and pulp and paper branches of industry during production of food additives, cleansers, medicinal drugs, stabilizers, etc. To obtain gluconic acid, the method of conversion of glucose into gluconic acid by molecular oxygen in the presence of solid catalysts is promising. The process of obtaining Pd and bimetallic nanoparticles Pd-Bi, coated on Al$_2$O$_3$, has been considered in the work. Samples were prepared by combined and successive impregnation of the Al$_2$O$_3$ support using metalloorganic precursors Pd(acac)$_2$, Bi(ac)$_3$, and dissolved in an organic solvent (acetic acid), followed by the removal of excess solvent. To achieve the formation of Pd and bimetallic nanoparticles Pd-Bi on the substrate surface, the synthesized samples were subjected to thermal decomposition sequentially in the atmosphere of Ar, O$_2$, and H$_2$. The surface of the obtained catalysts was studied by a combination of physicochemical methods of analysis. The catalysts were analyzed in the reaction of liquid phase oxidation of glucose. The best results are achieved in the presence of the catalyst obtained by combined impregnation.

**Keywords:** glucose oxidation; gluconic acid; bimetallic catalysts; palladium-bismuth; liquid phase oxidation

## 1. Introduction

Obtainment of gluconic acid and its derivatives is an important process of fine organic synthesis since these products are extensively used in different fields of industry [1,2]. In pharmaceutical production, gluconic acid and its salts are applied as a preservative, a buffer component to maintain the acidity of the medium of different cosmetic products, and in tableted dosage forms [3]. Calcium and iron gluconates are used in medicine to treat the deficiency of these trace elements by oral or intravenous administration. Zinc gluconate, due to its healing properties, is used as a component for treatment of wounds, colds, and infections [4]. Gluconic acid has an antioxidant action due to the presence of carboxyl and hydroxyl groups in the structure. In the food industry, gluconic acid is used in the form of a food additive as a conservative agent, leavening agent, and acidity regulator [5]. Because of the high complexing ability, gluconic acid salts are used as a chelating agent in detergents, in the manufacture of prostheses, in the leather, construction, and printing industries [6].

At present, gluconic acid is obtained by enzymatic processing of glucose in the presence of different microorganisms producing glucose oxidase: *Aspergillus niger,* some representatives of the *Penicillum* species, bacteria of *Pseudomonas*, yeast-like fungi *Aureobasidium pullulans*, acetic acid bacteria of *Acetobacter, Gluconobacter*, ectomycorrhizal fungi *Tricholoma robustum*, as well as a number of other microorganisms [6–17]. However, this method has a number of significant drawbacks: strict control of the composition of the nutrient medium, low fermentation rate, low volumetric capacity, formation of by-products, complexity of precipitation of the target product, impossibility of multiple application of enzymes, problems with waste disposal.

An alternative method of obtaining gluconic acid is aerobic oxidation of glucose in the presence of solid-phase nanodispersed systems based on active components—noble metals (Pd, Pt, Au) [18], promoted by a metal not involved in hydrogen sorption (Bi, Te, Co, Tl, etc.) and fixed on a support stable in an aqueous medium (C, SiO$_2$, Al$_2$O$_3$) [19–22]. Metal promoters change the electronic structure of the active component Pd, Pt, and Au, which can lead to a synergistic effect, increasing the selectivity of the target product yield.

Obtainment of gluconic acid in the presence of solid-phase catalysts has a number of advantages over enzymatic bioconversion of glucose: ease of isolation of the target product, repeated use of catalytic systems, high yields, and selectivity of gluconic acid formation. Using an environmentally safe water environment as a solvent eliminates the problems associated with waste utilization.

Despite the fact that bimetallic systems in the reaction of catalytic liquid-phase oxidation of glucose and other carbohydrates have been studied for a long time, effective methods for obtaining bimetallic particles of nanometer length scale have not been developed yet and the problems of fixing particles on the surface have not been solved. Bimetallic catalysts of the Pd-Bi composition are active not only in the reaction of glucose oxidation, but also in a number of other processes: oxidation of ethanol [23], formic acid [24], glyoxal [25], 1-phenylethanol [26], hydrodechlorination of 2.4-dichlorophenol [27], and water denitration [28]. Different methods of preparation of Pd-Bi bimetallic catalysts are known. In the work, Cai [23] and his colleagues used a method of subsequent treatment of the support (black activated carbon) with an aqueous solution of H$_2$PdCl$_4$ and Bi(NO$_3$)$_3$ in the presence of ethylene glycol as a stabilizing agent. After formation of Pd/C particles, there was irreversible adsorption of bismuth on their surface. Csilla Keresszegi and colleagues [26] obtained a Bi-Pd catalyst by depositing bismuth on the surface of the commercial catalyst Pd/Al$_2$O$_3$ in the presence of 2-propanol in the hydrogen atmosphere to maintain the reduced form of the metals. At the same time, Pd particles covered with adatoms and Bi layers were formed. In the work of Witońska [27], the catalysts of the Pd-Bi/SiO$_2$ and Pd-Bi/Al$_2$O$_3$ composition were prepared by successive impregnation of the support with, first, the PdCl$_2$ aqueous solution, then with the aqueous solution of Bi(NO$_3$)$_3$·5H$_2$O. The formation of fused structures PdBi and PdBi$_2$ was found. There is a method for preparing bimetallic Pd-Sn catalysts [29] for the process of butadiene hydrogenation by impregnating the Al$_2$O$_3$ support with solutions of metalloorganic precursors followed by heat treatment in the flow of Ar, O$_2$, and H$_2$. Such method of preparation of the catalysts allows obtaining nanoparticles containing atoms of two metals. We assumed that this method would be effective in forming bimetallic Pd and Bi particles that are active in the glucose oxidation reaction on the surface of Al$_2$O$_3$.

The purpose of this work is to study the influence of the method of preparation of the catalyst on the formation of Pd-Bi particles on the surface of Al$_2$O$_3$ and to research the catalysts in the reaction of glucose oxidation into gluconic acid.

## 2. Results and Discussion

In the course of obtaining the PdBi catalyst by the method of combined impregnation of the Al$_2$O$_3$ support with solutions Pd(acac)$_2$ and Bi(ac)$_3$, there was chemisorption of acetylacetonate due to the reaction with coordinately unsaturated sites of the support by the mechanism [30], leading to the formation of surface compounds of acetylacetone with aluminum of the [Al(acac)$_x$]$_s$ type. This mechanism was proposed for the reaction of interaction between the solution of the Pd(acac)$_2$

precursor and the surface of the $Al_2O_3$ support. We assumed that the interaction of the bismuth acetate solution with the $Al_2O_3$ surface would proceed similarly accompanied by the formation of surface-adsorbed aluminum acetate

$$Pd(acac)_2 + (Al^{3+})_s \rightarrow [Al(acac)_2]^+{}_s + (Pd^{2+})_s \tag{1}$$

$$Bi(ac)_3 + (Al^{3+})_s \rightarrow [Al(acac)_3]_s + (Bi^{3+})_s \tag{2}$$

During heat treatment of the catalyst in the argon flow (500 °C), the complex support, adsorbed by the surface, was decomposed accompanied by the formation of acetate, acetone and chemisorbed oxygen on the surface of $[Al–O]_s$ aluminum according to the reactions

$$[Al(acac)_2]^+{}_s \rightarrow [Al(Oac)_2]^+{}_s + 2CH_3–CO–CH_3 \tag{3}$$

$$[Al(Oac)_2]^+{}_s \rightarrow CH_3–CO–CH_3 + CO_2 + [Al–O]_s \tag{4}$$

$$[Al(acac)_3]_s \rightarrow CH_3–CO–CH_3 + CO_2 + [Al–O]_s \tag{5}$$

At the second stage of catalyst annealing in the oxygen flow (350 °C), acetone and metal surfaces were oxidized accompanied by the formation of chemisorbed particles of metal oxides $(PdO)_s$ and $(Bi_2O_3)_s$, carbon dioxide, and water

$$(Pd^{2+})_s + O_2 \rightarrow (PdO)_s \tag{6}$$

$$(Bi^{3+})_s + O_2 \rightarrow (Bi_2O_3)_s \tag{7}$$

$$CH_3\text{-}CO\text{-}CH_3 + O_2 \rightarrow CO_2 + H_2O \tag{8}$$

At the third stage of heat treatment of the catalyst in the hydrogen flow, the reaction of reduction of oxidized oxides PdO and $Bi_2O_3$, adsorbed on the support surface, to $Pd^0$ and $Bi^0$ proceeded

$$(PdO)_s + H_2 \rightarrow (Pd^0)_s + H_2O \tag{9}$$

$$(Bi_2O_3)_s + H_2 \rightarrow (Bi^0)_s + H_2O \tag{10}$$

During preparation of the Pd→Bi catalyst at the stage of impregnation of the support with the $Pd(acac)_2$ precursor solution, the same reactions proceeded as those for the PdBi catalyst (Equations (1), (3) and (4)). For the stage of impregnation of the support with a $Bi(NO_3)_3$ acidified solution, a mechanism of the interaction was proposed, by which a surface compound of nitrate with aluminum and bismuth was formed

$$Bi(NO_3)_3 + (Al^{3+})_s \rightarrow [Al(NO_3)_3]_s + (Bi^{3+})_s \tag{11}$$

Furthermore, at the stage of catalyst annealing (500 °C), the surface acetylacetonate complex (Equations (3) and (4)) and aluminum nitrate decomposed in the argon flow according to Equation (12)

$$[Al(NO_3)_3]_s \rightarrow [Al\text{-}O]_s + NO_2 + O_2 \tag{12}$$

During subsequent oxidation-reduction temperature treatments of the catalyst, the reactions proceeded according to Equations (6), (7), (9), (10).

According to the elemental analysis by microwave plasma atomic emission spectroscopy (MP-AES) (Table 1), it can be concluded that depending on the method of the catalyst preparation, a different amount of the active phase is applied to the support surface. The calculated content of the active phase was 1.5% mass. of Pd and 1.5% mass. of Bi (atomic ratio of Pd/Bi = 2). The Pd content on the surface of the monometallic catalyst determined by AES was 1.3%. The total content of the deposited components

(Pd + Bi) on the surface of bimetallic catalysts was 2.6%. According to the results of the elemental analysis, it is possible to note that with the same calculated ratio in the case of successive impregnation, bismuth is introduced in some excess, as compared to palladium, into the catalyst composition, which can lead to lower losses in Bi at the preparation stage. Thus, the content of active components of the catalyst prepared by successive impregnation is closest to the theoretically calculated one.

**Table 1.** Elemental analysis of obtained samples according to AES-MP data.

| Sample | Pd, % mass. | Bi, % mass. | n(Pd)/n(Bi) |
|---|---|---|---|
| $Pd/Al_2O_3$ | 1.3 | 0 | - |
| $Bi/Al_2O_3$ | 0 | 2.0 | - |
| $PdBi/Al_2O_3$ | 0.9 | 1.7 | 1.0 |
| $Pd{\rightarrow}Bi/Al_2O_3$ | 1.2 | 1.4 | 1.7 |

Studies of the surface of samples of catalysts $Pd/Al_2O_3$, $PdBi/Al_2O_3$, and $Pd{\rightarrow}Bi/Al_2O_3$ using transmission electron microscopy (TEM) reveals the dispersion of the metal and allows detecting the particle size distribution. TEM microphotographs illustrating the morphology of the samples supported on $Al_2O_3$ after the last stage of preparation (reduction at 500 °C during 2 h) are shown in Figure 1.

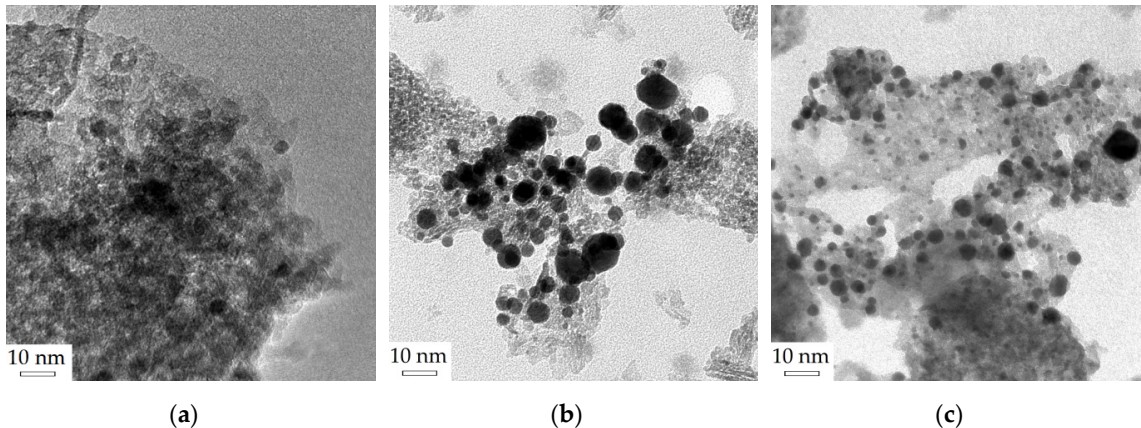

(**a**)        (**b**)        (**c**)

**Figure 1.** TEM-microphotographs of samples: (**a**) $Pd/Al_2O_3$; (**b**) $PdBi/Al_2O_3$; (**c**) $Pd{\rightarrow}Bi/Al_2O_3$.

For the $Pd/Al_2O_3$ monometallic catalyst, a narrower particle size distribution is observed as compared to samples of bimetallic catalysts. The particle sizes of the $Pd/Al_2O_3$ sample are in the range of 2–7 nm. For the $PdBi/Al_2O_3$ catalyst sample obtained by combined impregnation, small particles (0.5–16 nm) formation with a small contribution (>35 nm) of larger particles is observed. For the $Pd{\rightarrow}Bi/Al_2O_3$ sample prepared by sequential impregnation, a narrower particle size distribution (1–9 nm) is observed than that for the catalyst prepared by the method of combined impregnation.

Histograms of particle size distribution of all the catalysts after reduction at 500 °C in hydrogen during 2 h are shown in Figure 2.

When studying the materials by transmission electron microscopy with an energy dispersive spectrometer (TEM-EDS), it was found that in the case of the catalyst obtained by combined impregnation (Figure 3a,b), palladium and bismuth were located on the surface of the catalyst in close proximity to each other, which allows assuming an interaction between palladium atoms and bismuth atoms, leading to the formation of bimetallic particles. In the case of the catalyst prepared by successive impregnation, two types of particles are formed: mixed bimetallic Pd-Bi particles (Figure 4a) and monometallic Pd particles (Figure 4b).

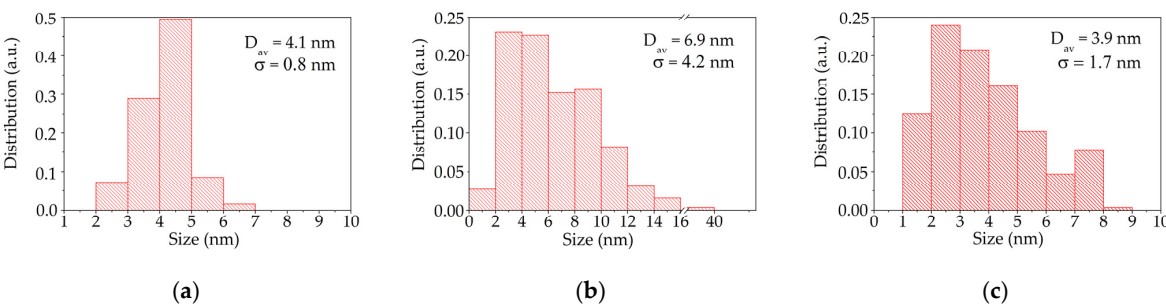

**(a)**　　　　　　　**(b)**　　　　　　　**(c)**

**Figure 2.** Historgrams of particle size distribution: (**a**) Pd/Al$_2$O$_3$; (**b**) PdBi/Al$_2$O$_3$; (**c**) Pd→Bi/Al$_2$O$_3$.

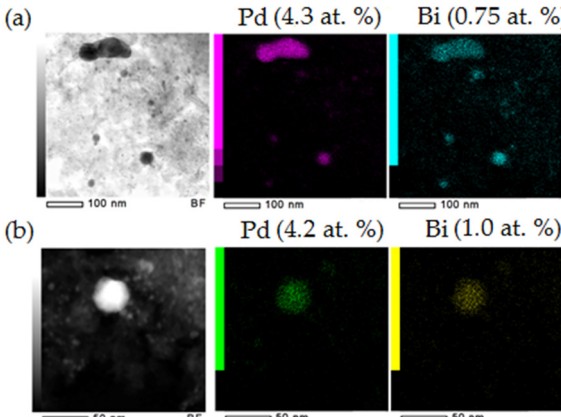

**Figure 3.** Elemental mapping of the surface of the PdBi/Al$_2$O$_3$ catalyst, obtained by EDS at (**a**) 400,000× magnification and (**b**) 1,000,000× magnification.

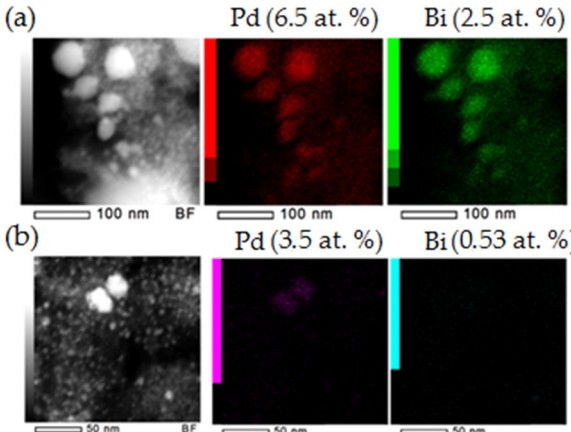

**Figure 4.** Elemental mapping of the surface of the Pd→Bi/Al$_2$O$_3$ catalyst by EDS at (**a**) 600,000× magnification and (**b**) 1,000,000× magnification.

Pointlike scanning of the sample surface of the PdBi/Al$_2$O$_3$ catalyst revealed the formation of particles with different atomic ratios of Pd/Bi (Figure 5). When scanning the surface of the Pd→Bi/Al$_2$O$_3$ catalyst, the formation of bimetallic PdBi particles with different atomic contents and monometallic Pd particles was also established (Figure 6). It should be also noted that the impurity content of bismuth Bi 1.0–5.0 at %. Was registered on the surface of some monometallic Pd particles. The atomic ratio of the metals at different scan points is presented in Table 2.

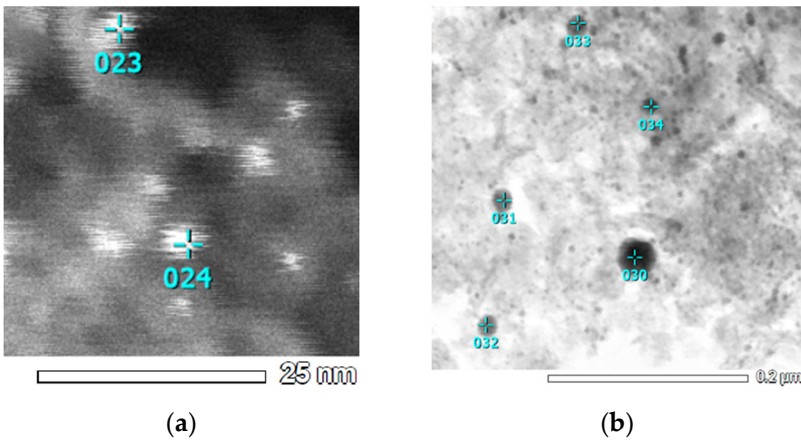

**Figure 5.** Pointlike scanning of the sample surface of the catalyst $PdBi/Al_2O_3$ by EDS at magnification of (**a**) 2,000,000× and (**b**) 400,000×.

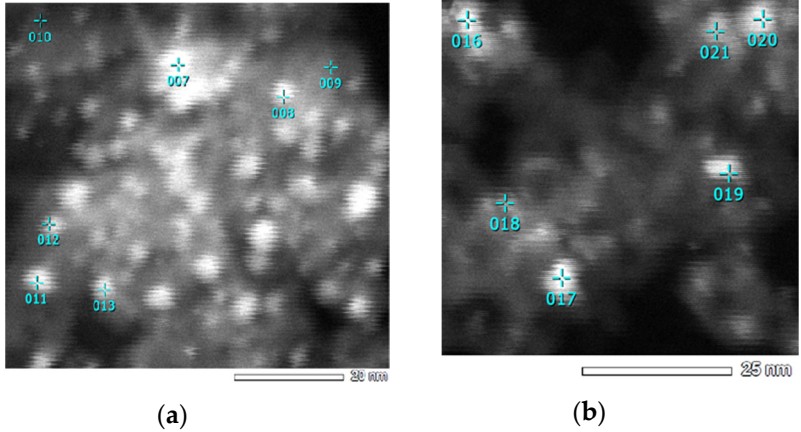

**Figure 6.** Pointlike scanning of the surface of the catalyst $Pd{\rightarrow}Bi/Al_2O_3$ by EDS at (**a**) 2,500,000× magnification and (**b**) 2,000,000× magnification.

**Table 2.** Atomic ratio of Pd/Bi at the scan points of the sample surface of the catalysts $PdBi/Al_2O_3$ and $Pd{\rightarrow}Bi/Al_2O_3$.

| No. of Points | At(Pd)/At(Bi) | No. of Points | At(Pd)/At(Bi) |
|:---:|:---:|:---:|:---:|
| 7 | 11.9 | 19 | 3.4 |
| 8 | Pd with impurity Bi | 20 | 3.5 |
| 9 | Pd with impurity Bi | 21 | 11.1 |
| 10 | Absence of Pd and Bi | 23 | 5.9 |
| 11 | Monometallic Pd | 24 | 1.5 |
| 12 | Absence of Pd and Bi | 30 | 2.4 |
| 13 | Pd with impurity Bi | 31 | 2.0 |
| 16 | 2.5 | 32 | 1.6 |
| 17 | 5.8 | 33 | 1.1 |
| 18 | 6.0 | 34 | 3.4 |

The surface composition and electronic properties for monometallic Pd and bimetallic catalysts Pd→Bi and PdBi were investigated by XPS. The XPS spectra for Pd 3d and Bi 4f of all the catalysts after reduction at 500 °C in a hydrogen flow during 2 h are shown in Figure 7. Table 3 shows the main results. Various chemical states of Pd can be analyzed by deconvolution of 3d-spectra of Pd. In the case of the $Pd/Al_2O_3$ sample, two asymmetric peaks differing by 5.3 eV, associated with orbitals Pd $3d_{5/2}$ and Pd $3d_{3/2}$, were found in the 3d spectrum of the palladium core (Figure 7a). Pd $3d_{5/2}$ signals can be

decomposed into two peaks: one with the center at 335.2 eV, defined for the metal form $Pd^0$ [31–34], and the second peak with the center at 336.4 eV, associated with the presence of an oxidized PdO phase [31–33,35–37]. For bimetallic catalyst samples, the binding energy of Pd $3d_{5/2}$ is in the range of 334.7–335.0 eV, which points to the presence of metallic forms of palladium. A shift in the peak of the binding energy of Pd $3d_{5/2}$ to the lower side for the Pd→Bi sample by 0.2 eV and 0.5 eV for the PdBi sample, respectively, was recorded, which is conditioned by the electronic interaction between palladium and bismuth [38]. For the binding energy of oxide forms of palladium Pd $3d_{3/2}$ in case of bimetallic catalysts, no displacements relative to the monometallic Pd sample are observed. However, in the binding energy region of PdO $3d_{3/2}$, there is a shift towards a higher binding energy (0.8 eV) for the Pd→Bi sample. The formation of PdO can be explained by a strong interaction of the precursor of the Pd(acac)$_2$ metal with the surface of the $Al_2O_3$ support.

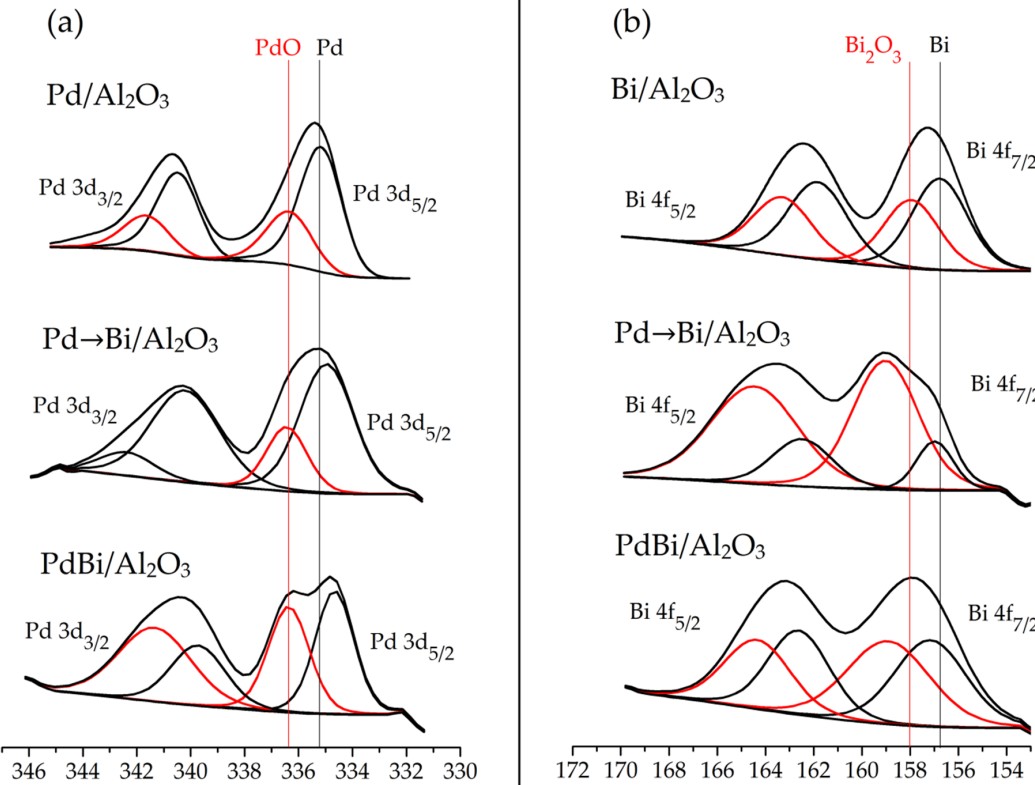

**Figure 7.** X-ray photoelectron spectra of sample surfaces of catalysts (**a**) Pd 3d, (**b**) Bi 4f.

**Table 3.** Binding energy (eV), atomic percent Pd, Bi and a proportion of metallic and oxide phases of catalyst samples for binding energies of Pd $3d_{5/2}$ and Bi $4f_{7/2}$.

| Sample | Pd $3d_{5/2}$ | Bi $4f_{7/2}$ | Pd, at % | Bi, at % | Proportion of $Pd^0$, % | Proportion of PdO, % | Proportion of $Bi^0$, % | Proportion of $Bi_2O_3$, % |
|---|---|---|---|---|---|---|---|---|
| Pd/Al$_2$O$_3$ | $Pd^0$ 335. 2 PdO 336.4 | - | 0.27 | | 68.8 | 31.2 | - | - |
| Bi/Al2O3 | - | $Bi^0$ 156.8 Bi$_2$O$_3$ 158.0 | 0.28 | | - | - | 57 | 43 |
| PdBi/Al2O3 | Pd0 334.7 PdO 336.4 | $Bi^0$ 157.3 Bi$_2$O$_3$ 159.0 | 0.34 | 0.39 | 53.8 | 46.2 | 47.1 | 52.9 |
| Pd→Bi/Al2O3 | Pd0 335.0 PdO 336.4 | $Bi^0$ 157.0 Bi$_2$O$_3$ 159.0 | 0.42 | 0.49 | 73.1 | 26.9 | 17.1 | 82.9 |

The XPS spectrum of monometallic catalyst Bi/Al$_2$O$_3$ consists of signals are $4f_{5/2}$ and $4f_{7/2}$. Bi $4f_{7/2}$ signals can be decomposed into two peaks: one with a center at 156.8 eV, attributed to the reduced form of the $Bi^0$ metal, and a second peak with a center at 158.0 eV, associated with the presence of the

oxidized form of $Bi_2O_3$. The binding energies in the region of 161.8 eV and 163.3 eV are associated with the reduced and oxide form of Bi $4f_{5/2}$ signals [31,39]. Deconvolution XPS spectra of Bi $4f_{5/2}$ and $4f_{7/2}$ of the core level for Bi, PdBi, and Pd→Bi are shown in Figure 8b. Both bimetallic samples demonstrate two similar components of bismuth species. For bimetallic PdBi and Pd→Bi samples in the Bi region in Figure 7b there are two doublets of binding energy-Bi $4f_{7/2}$ and Bi $4f_{5/2}$. The peaks in the region of 157.0–157.3 and 162.7 eV correspond to the reduced form of Bi [31]. However, there are shifts towards a higher binding energy (0.2–0.5 eV) relative to values of monometallic Bi catalyst, which confirms the presence of an interaction between palladium and bismuth. The binding energies in the region of 159.0 eV and 164.6 eV related to the oxide form of bismuth [31,39] also have displacements relative to the reference values. The presence of the Bi oxide form may be due to the remaining bismuth oxide, even if the catalysts were reduced in a hydrogen flow at 500 °C over 2 h. This is often the case because of the small amounts of the dead volume of oxygen present in the vacuum chambers of the X-ray photoelectron spectrometer when transferring the sample and because of the high propensity of Pd and Bi to oxidation during storage [23,38]. This observation, together with the fact that a part of the bismuth is reduced simultaneously with palladium using hydrogen temperature-programmed reduction (TPR-$H_2$), suggests, as will be shown below, that the rest of the unreduced bismuth is in bimetallic nanoparticles.

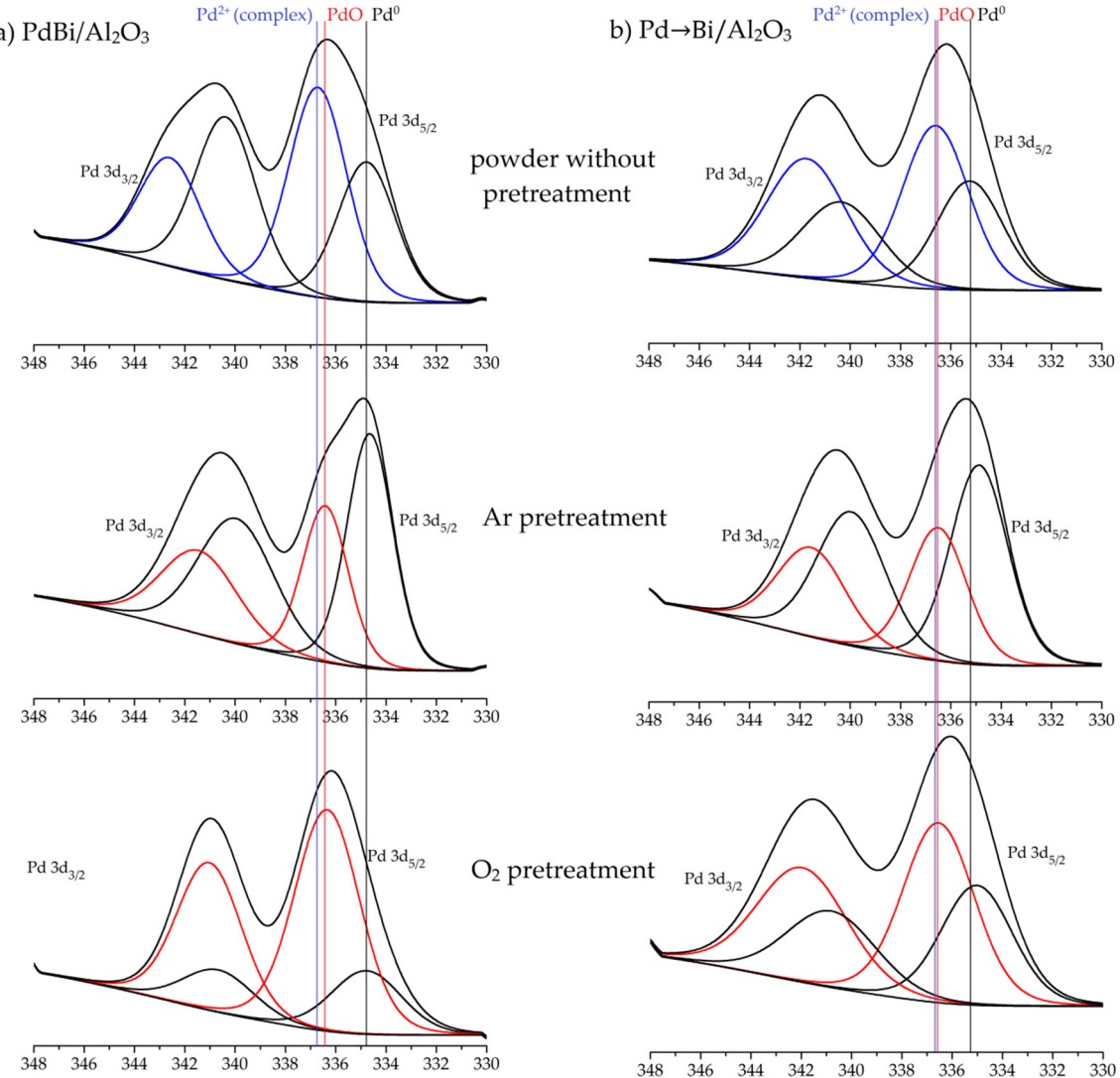

**Figure 8.** X-ray photoelectron spectra of the Pd 3d line of the catalyst surface after each stage of preparation (**a**) PdBi/$Al_2O_3$, (**b**) Pd→Bi/$Al_2O_3$.

The XPS surface analysis showed that after precipitation, one part of the coated forms of Pd(acac)$_2$ formed the PdO forms, the other part formed Pd$^0$ particles. The formation of PdO can be explained by a strong interaction of the metal precursor Pd(acac)$_2$ with the surface of Al$_2$O$_3$. Thermal decomposition in an inert Ar medium and subsequent reductive–oxidative treatments lead to an increase in the Pd$^0$ amount due to decomposition of acetylacetonate precursors and reduction of PdO. The results of the XPS-analysis show that there are two types of bismuth in Pd-Bi bimetallic samples corresponding to the reduced Bi$^0$ form and the oxidized Bi$_2$O$_3$ form. We cannot eliminate a probability of formation of the Pd-Bi-O compound phase.

The proportion of each metal phase on the surface of the studied samples was determined by the ratio of the peak areas of the reduced and oxidized Pd and Bi forms (Table 3). For the Pd→Bi sample, the proportion of the oxidized Bi form was higher (82.9%) as compared to the proportion of the oxidized form of the PdBi sample (52.9%). On the other hand, the proportion of the oxidized palladium form for the Pd→Bi sample (26.9%) was less than that for the PdBi sample (46.2%). This is due to the fact that in the case of obtaining the Pd→Bi catalyst by successive impregnation, bismuth is introduced after the impregnation stage by the Pa(acac)$_2$ precursor. In this case, the surface of palladium particles is coated with bismuth adatoms. Thus, bismuth is oxidized first, protecting the palladium surface from the oxygen influence. Preparation of the catalysts by combined impregnation contributes to a uniform distribution of palladium and bismuth particles. This leads to simultaneous oxidation of the surface of palladium and bismuth.

The X-ray photoelectron spectra of the catalysts surface were taken after each stage of preparation. The spectra of the Pd 3d and Bi 4f lines are shown in Figures 8 and 9, respectively.

The binding energy values of the Pd 3d, Bi 4f lines and the proportions of the phase states of metals are presented in the Table 4. At the stage of drying the catalyst powder in a vacuum oven, a peak of monometallic palladium Pd$^0$ is formed at a binding energy of 334.8–335.2 eV and Pd$^{2+}$ in the complex compound (336.6–336.7) in the form of Pd(acac)$_2$. The formation of monometallic palladium occurs during partial decomposition of the complex compound by evaporation of the solvent in a rotary evaporator or by drying the catalyst powder in a vacuum oven for 24 h at 80 °C. For bismuth, in the case of PdBi/Al$_2$O$_3$, a different picture is observed. A peak is formed with a center of 158.4 eV corresponding to the line of bismuth, which is in the form of Bi$^{3+}$, associated with bismuth acetate. In this case, Bi(ac)$_3$ partially decomposes with the formation of the oxide form Bi$_2$O$_3$ (159.1 eV). In the case of the catalyst obtained by sequentially impregnating the Pd→Bi/Al$_2$O$_3$ support, the peak of monometallic bismuth (157.0 eV) and Bi$^{3+}$ is formed in the form of Bi(ac)$_3$ (158.3 eV). At the stage of surface treatment of the catalysts with argon at 500 °C, both catalysts completely decompose salts of Pd(acac)$_2$ and Bi(ac)$_3$ with the formation of monometallic states (Pd$^0$ 334.6–334.9 eV, Bi$^0$ 158.7–159.0 eV) and oxides PdO (336.4 eV) and Bi$_2$O$_3$ (159.0 eV). At the last stage of preparation of the catalyst (surface treatment in a flow of hydrogen at 500 °C), the reduction of oxides to a metallic state does not occur completely (Figure 7). Apparently, in the case of the catalyst obtained by co-impregnation of PdBi/Al$_2$O$_3$, the formation of the interaction between palladium and bismuth begins at the stage of mixing organometallic precursors and drying in a vacuum oven.

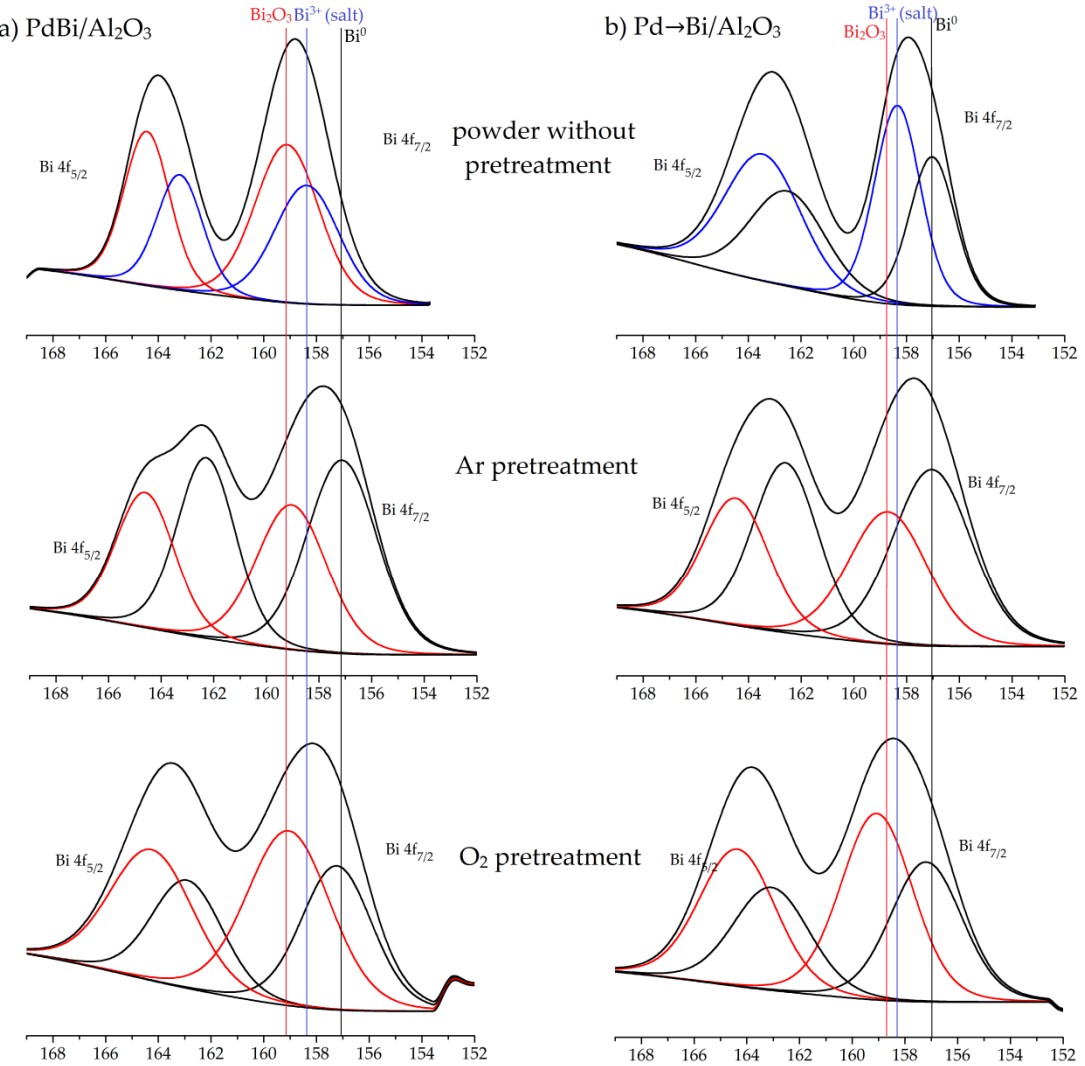

**Figure 9.** X-ray photoelectron spectra of the Bi 4f line of the catalyst surface after each stage of preparation (**a**) PdBi/Al$_2$O$_3$, (**b**) Pd→Bi/Al$_2$O$_3$.

**Table 4.** Binding energy (eV), atomic percentage of Pd, Bi, and the proportion of metal and oxide phases after each stage of catalyst preparation.

| Sample | Pd 3d$_{5/2}$ | Bi 4f$_{7/2}$ | Proportion of Pd$^0$, % | Proportion of Pd$^{2+}$, % | Proportion of Bi$^0$, % | Proportion of Bi$^{3+}$, % |
|---|---|---|---|---|---|---|
| PdBi/Al$_2$O$_3$ (without pretreatment) | Pd$^0$ 334.8 (Pd$^{2+}$)$_s$ 336.7 | Bi$^{2+}$ 158.4 Bi$_2$O$_3$ 159.1 | 40.0 | 60.0 | 42.9 | 57.1 |
| PdBi/Al$_2$O$_3$ (after Ar pretreatment) | Pd$^0$ 334.6 PdO 336.4 | Bi$^0$ 157.1 Bi$_2$O$_3$ 159.0 | 60.0 | 40.0 | 57.1 | 42.9 |
| PdBi/Al$_2$O$_3$ (after O$_2$ pretreatment) | Pd$^0$ 334.8 PdO 336.3 | Bi$^0$ 157.2 Bi$_2$O$_3$ 159.0 | 23.7 | 76.3 | 41.2 | 58.8 |
| Pd→Bi/Al$_2$O$_3$ (without pretreatment) | Pd$^0$ 335.2 (Pd$^{2+}$)$_s$ 336.6 | Bi$^0$ 157.0 (Bi$^{3+}$)$_s$ 158.3 | 40.0 | 60.0 | 42.9 | 57.1 |
| Pd→Bi/Al$_2$O$_3$ (after Ar pretreatment) | Pd$^0$ 334.9 PdO 336.5 | Bi$^0$ 157.0 Bi$_2$O$_3$ 158.7 | 60.0 | 40.0 | 57.1 | 42.9 |
| Pd→Bi/Al$_2$O$_3$ (after O$_2$ pretreatment) | Pd$^0$ 335.0 PdO 336.5 | Bi$^0$ 157.2 Bi$_2$O$_3$ 159.1 | 40.0 | 60.0 | 42.9 | 57.1 |

For the Pd→Bi/Al$_2$O$_3$ catalyst, the interaction between palladium and bismuth occurs at the stage of particle reduction in a stream of hydrogen. However, the shift in the binding energy of the Pd 3d and Bi 4f peaks of the finished Pd→Bi/Al$_2$O$_3$ catalyst relative to the Pd/Al$_2$O$_3$ monometallic catalyst is less

than for PdBi/Al$_2$O$_3$. This may mean different types of interaction between palladium and bismuth. PdBi/Al$_2$O$_3$ is characterized by the formation of bimetallic particles, while for PdBi/Al$_2$O$_3$, bimetallic forms with an admixture of monometallic particles are found.

The TPR-H$_2$ method was used to investigate the reductive behavior of Pd/Al$_2$O$_3$, PdBi/Al$_2$O$_3$, Pd→Bi/Al$_2$O$_3$, and Bi/Al$_2$O$_3$ samples. The obtained TPR profiles are shown in Figure 10. The sample of the Pd/Al$_2$O$_3$ monometallic catalyst demonstrated a TPR profile with one major reduction peak, which can be attributed to the reduction of PdO to Pd$^0$ metal [40]. The Pd/Al$_2$O$_3$ sample was reduced at 40 °C, while the reduction peak for both bimetallic samples shifted slightly to 53 °C. It is assumed that finer Pd particles and both samples of the bimetallic catalysts obtained from Pd(acac)$_2$ and Bi(ac)$_3$ precursors in acetic acid resulted in a stronger interaction with the Al$_2$O$_3$ support. The negative peak at 97 °C for the Pd monometallic catalyst sample can be explained by decomposition of the β-PdH phase, which is consistent with the literature data [41,42]. This negative peak of β-PdH decomposition indicates that some Pd particles formed β-hydride at the initial stages of reduction at low temperatures.

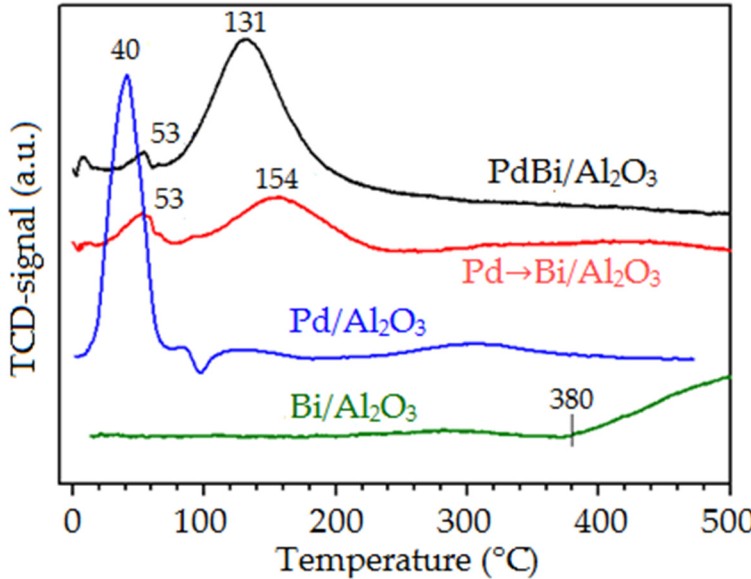

**Figure 10.** TPR profiles of all the synthesized samples: Bi/Al$_2$O$_3$, Pd/Al$_2$O$_3$, Pd→Bi/Al$_2$O$_3$, PdBi/Al$_2$O$_3$.

It is interesting to note that the Pd sample showed extended small reduction peaks in the temperature range of 250–350 °C, which may be due to the reduction of palladium nanoparticles strongly bound to the support surface. In contrast, both bimetallic samples showed no peak of β-PdH decomposition, which means that the presence of Bi limits the formation of this hydrogen-containing Pd phase [27,43]. Similar regularities were obtained for deposited catalysts promoted by Sn, Ag [44–46]. Instead, a positive peak was observed at 131–154 °C, which was observed for the samples PdBi/Al$_2$O$_3$ and Pd→Bi/Al$_2$O$_3$, but without a peak of β-PdH decomposition. This peculiarity is absent from the TPR profile of the Pd sample and is certainly related to the addition of Bi.

In the case of both Pd-Bi bimetallic samples, the intensity of the thermal conductivity detector signal (TCD signal) of PdO phase reduction decreased significantly along with the increasing Bi content (or decreasing Pd content). The appearance of an intensive peak in the range of 131–154 °C is probably due to the reduction the Pd-Bi-O compound phase. These observations allow suggesting that this unique peculiarity of TPR at 131–154 °C, present only in Pd-Bi bimetallic samples, may be related with mixed varieties of Pd-Bi, presumably the Pd-Bi alloy phase. This may mean that the presence of Bi inhibits the formation of β-PdH or that Pd is present in a different form such as the Pd-Bi alloy, which prevents from the formation of palladium β-hydride. Moreover, the intensity of this peak for PdBi is much higher than that for Pd→Bi, which indicates that the method of preparation of the catalysts influences the formation of Pd particles and the particles of the Pd-Bi compound phase.

This observation supports the idea that during combined preparation of the catalysts, the probability of obtaining the particles of the compound phase or fused structure of Pd-Bi is higher than that during a sequential introduction of Bi(ac)$_3$ after the Pd(acac)$_2$ precursor.

The beginning of the bismuth reduction is observed at 380 °C, and the volume reduction begins approximately at 500 °C [27]. It is interesting that the TPR profiles of the bimetallic catalysts show the absence of this peak. This peculiarity is associated with the formation of an interaction between palladium and bismuth, which leads to the suppression of the Bi$_2$O$_3$ reduction and appearance of the reduction peak of the mixed oxide phase Pd-Bi-O at 131–154 °C. A peculiarity of the TPR profile of the Pd→Bi sample prepared by successive impregnation is a flat extended peak of the Bi reduction above 300 °C. Peculiarities present in the temperature range of 300–450 °C have been attributed to the reduction of fine surface Bi varieties.

Based on the data of the elemental mapping and the study of the surface of PdBi/Al$_2$O$_3$, Pd→Bi/Al$_2$O$_3$ catalysts by XPS and TPR, the mechanism of formation of Pd and PdBi particles on the support surface was proposed (Figure 11).

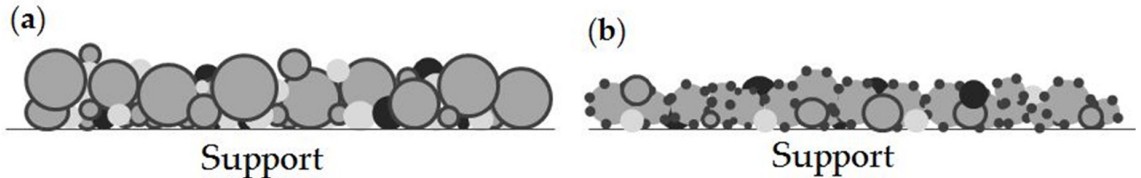

**Figure 11.** Mechanism of formation of particles on the surface of the support: (**a**) PdBi/Al$_2$O$_3$, (**b**) Pd→Bi/Al$_2$O$_3$.

In the course of preparation of the catalyst by combined impregnation, fine particles of palladium, bismuth, as well as coarser particles of PdBi, being in the interaction, are formed (Figure 11a). Preparation of the catalyst by successive impregnation, first, with a palladium precursor and then with a bismuth precursor, results in the formation of Pd particles coated with adatoms and layers of bismuth. A certain number of monometallic particles of Pd and Bi are also formed.

Catalysts PdBi/Al$_2$O$_3$, Pd→Bi/Al$_2$O$_3$ were studied in the reaction of glucose oxidation into gluconic acid at different molar ratios of 'glucose:palladium'. The ratios of n(Glucose)/n(Pd) for the PdBi/Al$_2$O$_3$ catalyst were 1:600, 1:1900, 1:3800, 1:5000. For the Pd→Bi/Al$_2$O$_3$ catalyst the following ratios of n(Glucose)/n(Pd) were chosen: 1:500, 1:1400, 1:2900, 1:5000. Table 5 presents data on glucose (X) conversion, gluconic acid selectivity (S), gluconic acid (Y) yield, turnover numbers (TONs), and turnover frequencies (TOFs) within 110 min of reaction on synthesized catalysts at different ratios of the amount of glucose substance in relation to the amount of palladium substance (nGlucose/nPd). The kinetic dependence of glucose conversion over time is shown in Figure 12. After catalytic tests, the surface of the catalysts was studied by XPS (Figure 13).

**Table 5.** Kinetic parameters of the reaction.

| Sample | n (Glucose)/n (Pd) | X (Glucose), % | S (Gluc. Acid), % | Y (Gluc. Acid), % | TONs | TOFs, min⁻¹ |
|---|---|---|---|---|---|---|
| PdBi/Al$_2$O$_3$ | 600:1 | 100 | 83.0 | 83.0 | 1104 | 10.0 |
| | 1900:1 | 100 | 95.5 | 95.5 | 3495 | 25.0 |
| | 3800:1 | 95.2 | 81.1 | 77.2 | 6655 | 60.5 |
| - | 5000:1 | 83.7 | 99.9 | 83.6 | 7700 | 70 |
| Pd→Bi/Al$_2$O$_3$ | 500:1 | 100 | 65.0 | 65.0 | 998 | 9.1 |
| | 1400:1 | 100 | 84.2 | 84.2 | 2796 | 25.4 |
| | 2900:1 | 100 | 84.1 | 84.1 | 5791 | 52.6 |
| - | 5000:1 | 63.7 | 97.0 | 61.8 | 6357 | 57.8 |

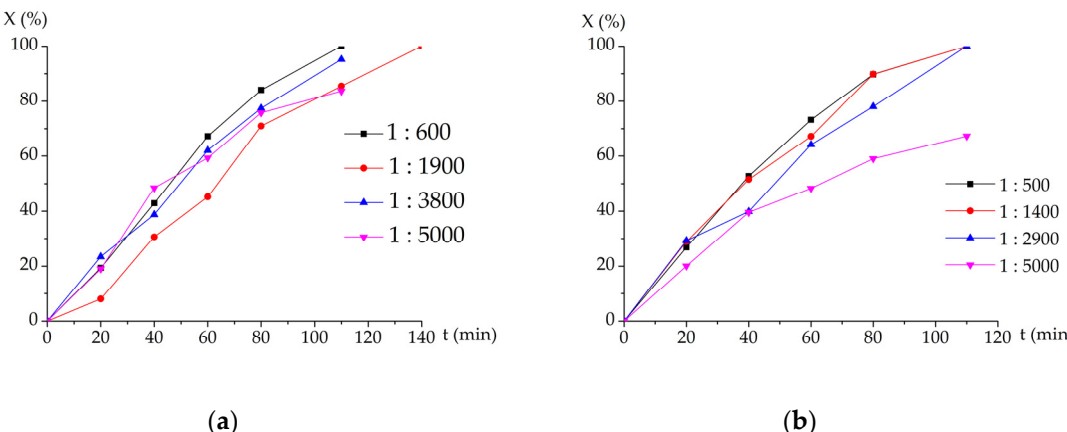

(**a**)           (**b**)

**Figure 12.** Dependence of glucose conversion (X) on time (t) in the presence of the catalyst (**a**) PdBi/Al$_2$O$_3$ (**b**) Pd→Bi/Al$_2$O$_3$.

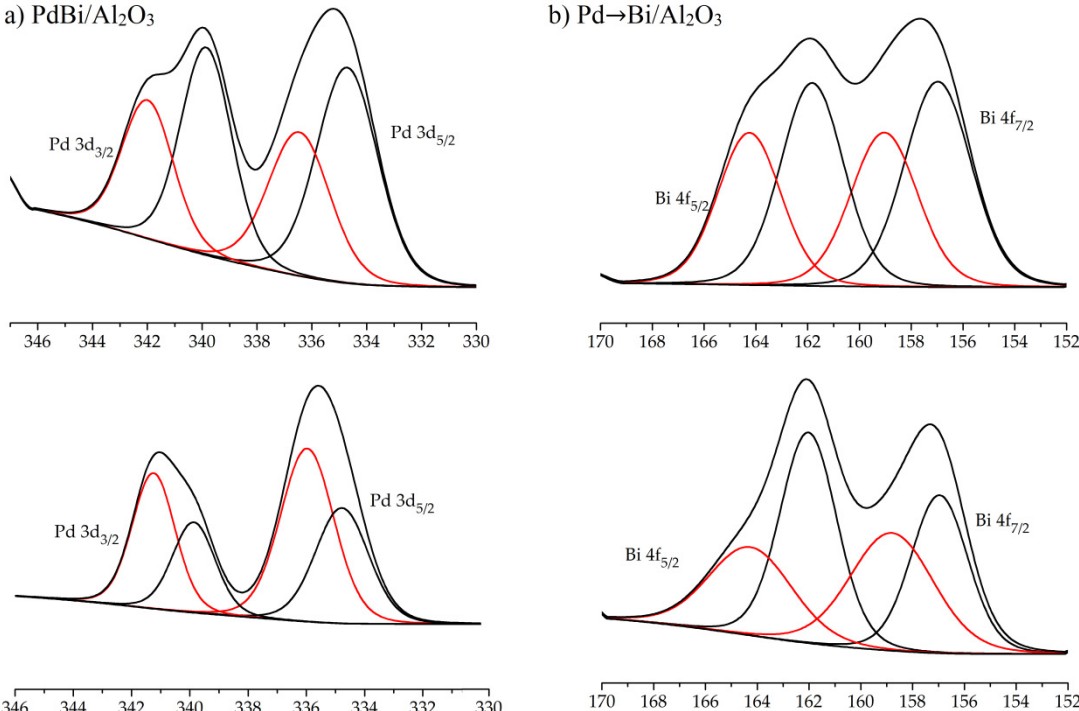

**Figure 13.** X-ray photoelectron spectra of the surfaces of catalyst samples after catalytic tests (**a**) PdBi/Al$_2$O$_3$, (**b**) PdBi/Al$_2$O$_3$.

During catalytic oxidation, in addition to glucose and gluconic acid, compounds such as fructose and glucaric acid were identified. The highest values of the yield of the target product and selectivity for gluconic acid were achieved in the presence of a catalyst prepared by combined impregnation of PdBi/Al$_2$O$_3$, at the molar ratio 'glucose:catalyst' = 5000:1. At a given molar ratio in the presence of a catalyst PdBi/Al$_2$O$_3$, a higher glucose conversion (83.7%) is achieved compared to Pd→Bi/Al$_2$O$_3$ (67.3%). In incomplete glucose conversion, fructose is the only by-product. It can be noted that the binding energies in the XPS spectra of the catalyst surfaces did not change (Table 6).

**Table 6.** Binding energy (eV), atomic percent Pd, Bi and a proportion of metallic and oxide phases of catalyst samples for binding energies of Pd 3d$_{5/2}$ and Bi 4f$_{7/2}$ after catalytic tests.

| Sample | Pd 3d$_{5/2}$ | Bi 4f$_{7/2}$ | Proportion of Pd$^0$, % | Proportion of PdO, % | Proportion of Bi$^0$, % | Proportion of Bi$_2$O$_3$, % |
|---|---|---|---|---|---|---|
| PdBi/Al$_2$O$_3$ | Pd$^0$ 334.7 PdO 336.5 | Bi$^0$ 156.9 Bi$_2$O$_3$ 159.0 | 60.0 | 40.0 | 57.1 | 42.9 |
| Pd→Bi/Al$_2$O$_3$ | Pd$^0$ 334.6 PdO 335.8 | Bi$^0$ 156.9 Bi$_2$O$_3$ 158.8 | 40.0 | 60.0 | 47.8 | 52.2 |

In the case of the PdBi/Al$_2$O$_3$ catalyst, the proportions of the metal and oxide phases were preserved. However, despite the high selectivity of the process for the Pd→Bi/Al$_2$O$_3$ catalyst at a molar ratio of 1:5000, the prevalence of the oxide phases PdO and Bi$_2$O$_3$ is observed in the X-ray photoelectron spectra (more than 50%). This is due to the fact that during preparation of the catalyst by combined impregnation bimetallic structures are produced, where bismuth is less oxidized as compared to the catalyst sample Pd→Bi/Al$_2$O$_3$. Bismuth in the bimetallic particle PdBi improves the properties of the catalyst, preventing from oxidation of the active component and providing high values of the target product yield and selectivity, according to the mechanism of oxidative dehydrogenation proposed by Besson and his colleagues (Figure 14) [47].

**Figure 14.** Glucose oxidation mechanism in the presence of the palladium-bismuth catalyst.

In this case, oxygen is adsorbed on bismuth accompanied by the formation of Bi$_2$O$_3$. Bismuth oxide reacts with hydrogen, splits off from glucose, is adsorbed on the palladium surface, and turns into water, protecting the catalyst surface from oxidation. Synthesis of the catalysts by successive impregnation of Pd→Bi/Al$_2$O$_3$ leads to the formation of particles of the mixed type (bimetallic and monometallic structures). In this case, the proportion of bismuth in the oxidized state is significant. Bismuth in the oxidized form is able to a lesser degree to protect the surface of the active component from oxidation, which leads to lower yields of the target product and the formation of accessory substances. As full glucose conversion is achieved, there is an increase in the yield of glucaric acid is the product of subsequent oxidation of gluconic acid and fructose is the product of glucose isomerization.

## 3. Materials and Methods

### 3.1. Preparation of the $Al_2O_3$ Support

We have chosen $Al_2O_3$ obtained from gibbsite by centrifugal thermal activation followed by acid hydration under mild conditions [48]. Based on the decomposition temperatures and weight loss adopted for hydroxides, their percentage in hydration products was calculated. The results of differential thermal analysis (DTA) are presented in Table 7. The data indicate the possible presence of various phases in the analyzed sample (boehmite, pseudoboehmite, amorphous phase). The acid hydration product in terms of water content corresponds to pseudoboehmite (n = 1.54). The obtained alumina contains boehmites of varying degrees of crystallization: well-crystallized boehmite (decomposition temperature 511 °C), pseudoboehmite (decomposition temperature 434 °C), and amorphous oxyhydroxide (decomposition temperature about 300 °C).

**Table 7.** Results of differential thermal analysis.

| DTA Results, % | | | n, $Al_2O_3 \cdot nH_2O$ |
|---|---|---|---|
| **boehmite** | **pseudoboehmite** | **amorphous phase** | 1.54 |
| 30.7 | 38.7 | 30.6 | |

The specific surface area of the $Al_2O_3$ support was determined by physical adsorption of $N_2$ using an automated system Micromeritics TriStar 3020 (Micromeritics, Norcross, GA, USA). Before measurements, about 100 mg of samples were placed in cylindrical reactors and subjected to thermal vacuum treatment at the VacPrep 061 (Micromeretics, Norcross, GA, USA) degassing station (pressure 50–100 mTorr) at a temperature of 200 °C with a gradient of 5 °C/min for 120 min to remove residual contaminants from the surface. samples. Then, the reactors with the samples were placed in the analyzer, where the adsorption–desorption isotherm was automatically constructed over the entire range of relative pressure of the adsorbate $p/p_s$. The Brunauer-Emmett-Teller (BET) method was used to estimate the specific surface area, and the Barrett-Joyner-Halenda (BJH) method was used to determine the pore volume. The measured total specific surface area ($S_{BET}$) was 301 m$^2$/g and the pore volume was 0.37 cm$^3$/g. Physico-chemical characteristics of the obtained alumina are presented in Table 8. Cylindrical alumina granules were ground to produce a very fine powder. The ground powder was dried in a vacuum cabinet at 120 °C for 24 h. Then it was sifted and the fraction of <70 μm was selected.

**Table 8.** Physico-chemical properties of $Al_2O_3$

| Total $S_{BET}$, m$^2$/g | $S_{BET}$ <1.7 nm, m$^2$/g | $S_{BET}$ <1.7–300 nm, m$^2$/g | Total $V_{pore}$ ($N_2$), cm$^3$/g | $V_{pore}$ ($N_2$) 1.7–300 nm, cm$^3$/g | $V_{pore}$ ($N_2$) <1.7 nm, cm$^3$/g | The Average Pore Size D, Å | Mechanical Strength, MPa | Bulk Density, g/cm$^3$ |
|---|---|---|---|---|---|---|---|---|
| 301 | 0 | 242 | 0.37 | 0.28 | 0 | 48.5 | 7.1 | 0.802 |

### 3.2. Synthesis of Catalysts

Two monometallic $Pd/Al_2O_3$ and $Bi/Al_2O_3$ and two bimetallic $PdBi/Al_2O_3$ and $Pd{\rightarrow}Bi/Al_2O_3$ catalysts were synthesized in the work. The $PdBi/Al_2O_3$ catalyst was obtained by combined impregnation of the support with precursors' solutions; the $Pd{\rightarrow}Bi/Al_2O_3$ catalyst was obtained by successive impregnation of the support with precursors' solutions. Palladium was chosen as an active component in the glucose oxidation reaction. Bismuth was chosen as a promotional metal. The atomic ratio between the active metal and the Pd:Bi metal-modifier was 2:1. The calculated content of each component was 1.5 mass %, the total amount of the supported metals was 3 wt. %. The catalysts were synthesized in two ways.

The $PdBi/Al_2O_3$ catalysts were obtained by combined diffusion impregnation of the support with precursors' solutions. $Pd(acac)_2$ (Sigma-Aldrich, Saint Louis, MO, USA, 99%) and $Bi(Ac)_3$

(Sigma-Aldrich, Saint Louis, MO, USA, 99.99%) were dissolved in excess of glacial acetic acid (Ecos-1, Staraya Kupavna, Moscow region, Russia, 99.5%). Then the support was added to the solution and allowed to stir for 18 h. Using a rotary evaporator in a water bath (55 °C, 60–80 rev./min.), acetic acid was driven to dry. The resulting powder was dried in the vacuum cabinet at a temperature of 80 °C for 24 h and then loaded into the quartz reactor with a membrane. The reactor containing the powder was placed in the tubular thermoprogrammed furnace and a sequential three-stage treatment was performed: (1) the oven was heated in the argon flow (60 mL/min) to 500 °C at a rate of 1 °C /min, held for 2 h, then cooled; (2) the furnace was heated in the oxygen flow (60 mL/min) to 350 °C at a rate of 1 °C/min, held for 2 h, then cooled; (3) the furnace was heated in the hydrogen flow (60 mL/min) to 500 °C at a rate of 1 °C /min, held for 2 h, then cooled. Argon (gas purity 99.7%) and oxygen (99.99%) were supplied from a gas cylinder (OOO 'GVS', Tomsk, Tomsk region, Russia). Hydrogen was obtained by the electrolysis of $H_2O$ using a pure hydrogen generator QL-300 (purity 99.99%).

Catalysts Pd→Bi/$Al_2O_3$ were prepared by successive impregnation of the support from the solutions of the precursors of different nature. Pd(acac)$_2$ (Sigma-Aldrich, USA, 99%) was dissolved in excess of toluene (Ecos-1, Staraya Kupavna, Moscow region, Russia, 99.5%). Then the support was added to the solution and allowed to stir for 18 h (diffusion impregnation). Using the rotary evaporator in a water bath (50 °C, 60–80 rev./min), toluene was driven to dry. The resulting powder was dried in the vacuum cabinet at a temperature of 80 °C for 24 h. Bi(NO$_3$)$_3$·5H$_2$O (Sigma-Aldrich, USA, 98%) was dissolved in the solution of nitric acid with pH 3 and applied to the support powder with fixed Pd impregnation by moisture capacity and then allowed to soak for 5 h. The powder with the catalyst was placed in the vacuum drying cabinet at a temperature of 80 °C for 24 h. The dried powder was loaded into the quartz reactor with a membrane. The reactor containing the powder was placed in the tubular thermoprogrammable furnace. Then three stages of heat treatment with argon, oxygen and hydrogen followed, similar to the method of catalyst preparation by combined impregnation.

The monometallic Pd/$Al_2O_3$ catalyst was synthesized by method 2, excluding the stage of impregnation with a solution of five-water bismuth nitrate. The monometallic Bi/$Al_2O_3$ catalyst was prepared by incipient wetness impregnation with the solution of five-water bismuth nitrate.

According nitrogen adsorption, the BET specific surface area and pore volume after preparation of the catalyst decreases compared with the initial support. The values of the total specific surface area ($S_{BET}$) and pore volume ($V_{pore}$) are shown in Table 9.

**Table 9.** Total surface area ($S_{BET}$) and total pore volume ($V_{pore}$) of obtained catalysts according to nitrogen adsorption

| Sample | Total Specific Surface Area ($S_{BET}$), $m^2/g$ | Total $V_{pore}$ ($N_2$), $cm^3/g$ |
|---|---|---|
| Pd/$Al_2O_3$ | 239 | 0.32 |
| PdBi/$Al_2O_3$ | 239 | 0.32 |
| Pd→Bi/$Al_2O_3$ | 240 | 0.32 |

### 3.3. Atomic Emission Spectrometry with Microwave Plasma (MP-AES)

The content of the quantitative composition of active components on the surface of the catalyst was determined by the MP-AES method on the Agilent 4100 device (Agilent Technologies, Santa Clara, CA, USA). Sample preparation was carried out by dissolving $HNO_3$ (LenReactiv, St. Petersburg, Russia, 65%) + HF (SigmaTec, Khimki, Moscow region, Russia, 45%) in the mixture. The state standard reference samples Pd and Bi were used as standards for spectrometer calibration.

### 3.4. Transmission Electron Microscopy with Energy Dispersive Spectrometer (TEM-EDS)

The morphology, particle size distribution and local chemical composition of nanoparticles were determined by transmission electron microscopy (TEM) and energy dispersive spectrometry (EDS). We used JEOL JEM-2100F (JEOL Ltd., Akishima, Tokyo, Japan) operating at 200 kV, equipped with

an electron gun with field emission of the cathode (FEG), a high-resolution pole tip (with a point resolution of 0.19 nm) and a JEOL JED-2300 Analysis Station spectrometer (JEOL Ltd., Akishima, Tokyo, Japan). The samples were suspended in ethanol, treated with ultrasound and dispersed on the electron microscopic lattice Cu (3.05 mm, 300 meshes) covered with a film made of leaky carbon before observation. The overall morphology of the samples, along with the distribution of particles of Pd and PdBi nanoparticles, was obtained by means of conventional TEM observations. The local structure can be obtained using high-resolution visualization (HRTEM).

### 3.5. X-ray Photoelectronic Spectroscopy

The surface of the catalysts was studied by XPS. The measurements were carried out using a 100-micron X-ray beam on a PHI 5000 VersaProbe-II (ULVAC-PHI, Chigasaki, Kanagawa, Japan) instrument equipped with argon and electron guns, which were used to neutralize the charge arising in the analysis of non-conducting samples (double beam charge neutralization method). The Al 2p line was taken as an internal standard at 73.4 and 73.6 eV. The accuracy of measurements of the binding energy was ± 0.1 eV for all the samples. The deconvolution of XPS peaks was carried out by a mixed Gaussian–Lorentzian correspondence by simultaneously subtracting the background caused by secondary electrons and photoelectrons losing energy, according to the Shirley algorithm. The XPS spectra were processed using standard CasaXPS software (Version 2.3.22PR1.0, 2018, Casa Software Ltd., Wilmslow, Cheshire, United Kingdom).

### 3.6. Temperature-Programmed Reduction with Hydrogen (TPR-$H_2$)

The reactivity of the samples in relation to $H_2$ was studied by the temperature-programmed reduction of TPR-$H_2$. TPR-$H_2$ measurements were made using the automated system of the chemisorption analyzer AutoChem 2950 HP (Micromeretics, Norcross, GA, USA). Prior to the analysis, the catalysts were oxidized in the $O_2$ flow at 350 °C for 10 min at a heating rate of 10 °C/min. A 50 mg catalyst sample, the temperature rising from −50 to 500 °C at a rate of 10 °C/min, and a gas-carrier consisting of 10% $H_2$ in Ar were used.

### 3.7. Research of Catalytic Properties

Liquid phase oxidation of glucose was made at 60 °C, atmospheric pressure and pH of 8.8–9.2 in a three-necked glass reactor. The pH was maintained by a peristaltic pump by supplying alkali, control was carried out using a glass combined electrode throughout the entire catalytic process. Glucose suspension (3.1 g) was previously dissolved in the reactor in 25 mL of water. The initial glucose concentration was approximately 0.6 mol/l. Then the calculated amount of the catalyst was added, pH was adjusted to 9 by adding the required amount of 3M NaOH (LenReactiv, St. Petersburg, Leningrad region, Russia, 99%) [49], oxygen was supplied to the suspension at a flow rate of 10 mL/min. The reaction was carried out for 110 min with continuous stirring (1500 rev/min). The samples for the analysis were periodically taken from the reaction mixture. Catalytic parameters of the catalysts after the catalytic reaction—such as glucose conversion (X), gluconic acid selectivity (S), and gluconic acid yield (Y)—were calculated using the formulas presented in [20].

### 3.8. HPLC Analysis of Catalytic Reaction Products

The method for fast determination of glucose, potential products of glucose oxidation (gluconic, glucuronic, and glucaric acids) and isomerization (fructose) is essential to evaluate conversion and selectivity of the reaction. Analysis of carboxylic acids is typically performed on ion-exchange resins with diluted sulfuric acid as mobile phase and UV-detection at 210 nm [50,51]. However, glucose and its isomerization products are characterized with a low UV-absorption [52]. Here, the refractive index detector was used. We found that a good resolution between peaks of glucose and gluconic acid cannot be achieved in ion-exchange mode [50].

We developed new chromatographic conditions for simultaneous determination of carbohydrates and organic acids on aminopropyl silica gel stationary phase and acetonitrile-phosphate buffer mixture as mobile phase. A mixed-mode of hydrophilic interaction and ion chromatography was carried out. The effect of IC mechanism was controlled by pH of the buffer, while buffer–acetonitrile ratio-controlled glucose, fructose, and organic acids retention. The optimal conditions were following: refractive index detector; column Zorbax NH2 250 × 4.6 mm, 5 μm (Agilent, USA); column temperature +55 °C; flow rate—3 mL/min; mobile phase–phosphate buffer solution (pH = 2.5)–acetonitrile in a ratio of 1:4; injection volume—100 μL. Chromatogram of the model solution of D-glucose (Sigma-Aldrich, 99.5%), D-fructose (Sigma-Aldrich, 99.0%), sodium salt of D-gluconic acid (Sigma-Aldrich, 99.0%) and D-glucuronic acid (Sigma-Aldrich, 98.0%) in the optimal conditions is shown on the Figure 15.

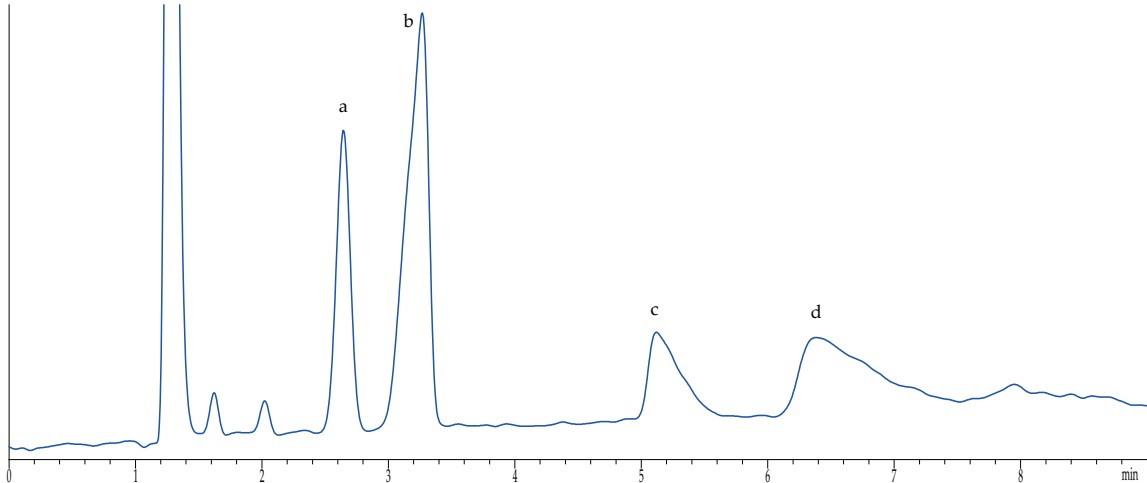

**Figure 15.** Chromatogram of the model solution of D-glucose (**a**), D-fructose (**b**), sodium salt of D-gluconic acid (**c**), and D-glucuronic acid (**d**).

Glucaric acid cannot be determined by this method due to the high retention. Hence, the absence of glucaric acid in the samples was preliminary shown with HPLC on Rezex ROA organic acid column. To increase the sensitivity of the method the sample preparation procedure was proposed. Aliquot of 700 μL of the reaction mixture were transferred into the Eppendorf tube and cooled down to 0–5 °C to stop the reaction. The sample was centrifuged at 15,000 rpm for 1 min. 200 μL of supernatant were transferred into a vial and added with 800 μL of acetonitrile for HPLC. The resulting solution was shaken well before HPLC analysis.

The developed HPLC method combined with the proposed sample preparation procedure provides the determination of glucose and fructose over the range 2 to 150 mg/mL, gluconic acid and other potential by-products 5 to 150 mg/mL in the reaction mixture within 10 min at isocratic flow.

## 4. Conclusions

The PdBi/Al$_2$O$_3$ and Pd→Bi/Al$_2$O$_3$ catalysts were prepared by the methods of co-impregnation and sequential impregnation of the support. In the presence of both catalysts, high selectivities were achieved (99.9% for PdBi/Al$_2$O$_3$ and 97.0% for Pd→Bi/Al$_2$O$_3$, respectively) in the oxidation of glucose to gluconic acid with incomplete glucose conversion (molar ratio of glucose: catalyst = 5000: 1). Investigation of the catalyst samples surface before and after catalytic tests shows that the catalyst Pd→Bi/Al$_2$O$_3$ obtained by sequential impregnation is oxidized after the catalytic test because bismuth was initially in the oxidized state and did not lead to the promoting effect of palladium. The catalyst PdBi/Al$_2$O$_3$ obtained by the method of co-impregnation preserves the initial state of the surface. Bismuth prevents the oxidation of the active component (palladium) by adsorbing oxygen on

its surface, which interacts with hydrogen adsorbed on the surface of palladium, turning into water. These transformations lead to the restoration of the surface of the catalyst.

**Author Contributions:** M.P.S.—Conceptualization, Data curation, Formal analysis, Investigation, Methodology, Validation, Visualization, Writing—original draft, Writing—review & editing. V.S.S.—Formal analysis, Investigation, Methodology, Validation, Writing—original draft. A.A.G.—Methodology, Software, Investigation. A.V.C.—Methodology, Software, Investigation. I.A.K.—Conceptualization, Data curation, Formal analysis, Funding acquisition, Project administration, Resources, Software, Supervision, Visualization All authors have read and agreed to the published version of the manuscript.

**Funding:** This research was financially supported by competitiveness improvement program of The National Research Tomsk State University (grant no. 8.2.10.2018).

**Acknowledgments:** The authors accord a thank Daniil A. Zuza for studies of catalytic performances in glucose oxidation reaction.

**Conflicts of Interest:** The authors declare no conflict of interest.

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
