# Peer review of "Influence of the Method of Preparation of the Pd-Bi/Al2O3 Catalyst on Catalytic Properties in the Reaction of Liquid-Phase Oxidation of Glucose into Gluconic Acid"

_catalysts, doi:10.3390/catal10030271_

Round 1
Reviewer 1 Report
The liquid phase oxidation of glucose, with molecular oxygen, in the presence of promoted noble metal (Pd and Pt) based catalysts has been extensively studied in the last decade. Also, it is well known that bismuth is the most common promoter. The present paper is another variation in this field. In this report, what is the new concept highlighted to identify it compared to others (J. Molecular Catalysis A, 180, 2002. 141; Applied Catalysis A, 497, 2015, 22; International J. Hydrogen Energy, 28, 2014, 15855)? However, I do not find in-depth results and clear evidences for a novelty and a significant advance. Also, the obtained data quality (especially TEM images) is very low. Therefore I recommend rejection.
Author Response
Dear Reviewer,
thank you very much for the reviewing of our manuscript titled as Influence of the method of preparation of the Pd-Bi/Al2O3 catalyst on catalytic properties in the reaction of liquid-phase oxidation of glucose into gluconic acid. We have made a number of significant changes to the manuscript, and now we believe that it can be published in the journal Catalysts.
Sincerely,
Irina A. Kurzina, Prof.
National Research Tomsk State University
Director of the Centre for Research in
Materials and Technologies
Tel: +7-9138821028
Reviewer 2 Report
Dear Authors,
In my opinion, your manuscript titled: “Influence of the method of preparation of the Pd-Bi/Al2O3 catalyst on catalytic properties in the reaction of liquid-phase oxidation of glucose into gluconic acid” can be published in Catalysts after major revision.
Please, you can find below remarks, suggestions and comments to your work.
In the case of presented samples, in my opinion, you should describe Al2O3 as a support, because the addition of Bi and Pd did not change the main physicochemical properties of Al2O3. It is difficult to estimate this because you don’t show the characterisation data of the support. Of course, the word “a carrier” is sometimes applied to the description of catalysts for the chemical industry.
You have to add the description of sample pre-treatment before the adsorption and desorption of nitrogen (the amount of sample, the temperature heating rate, the method which was applied to estimate the specific surface area, the total pore volume, etc.).
You have to add the information about the producer and quality of gasses (O2, Ar, H2) and chemical compounds (NaOH, HNO3, HF) which were applied to the analysis or the catalytic tests.
You have to add the information about the temperature heating rate which was used to the pre-treatment of catalysts in the temperature-programmed reduction of samples by hydrogen (from RT(?) to 350oC).
You applied two bands Al species for the standardization of XP spectra. Please, could you explain, did you always apply two bands (at 73.4 and 73.6 eV) or did you choose only one for the selected group of XP spectra?
You have to add the description of the deconvolution process which was applied for XP spectra.
Additionally, in my opinion, you have to record the XP spectra for the catalysts after their oxidation by oxygen and the reduction by hydrogen at a selected temperature (at around 100oC, 150oC, 350oC and 500oC). This study can explain the possible changes in the chemical composition of catalysts, especially on the eternal surface.
You have to correct of selected Figures, e.g. in Figures 1-6, the quality of TEM images is too poor. The size of the particles is difficult to estimate. In Figure 9, you have to remove the error in the description and the graph presented the changes in the support is illegible. You have to add the graph with the results in Figure 10b.
You have to calculate the values of TOFs and/or TONs achieved over all catalysts and for each molar ratio glucose/Pd.
You have to perform the characterisation of materials after catalytic test using, e.g. TEM and/or XPS. The studies of materials characterisation can confirm the potential changes in the catalysts.
You have to prepare Bi/Al2O3 sample and this sample should be characterised using X-ray photoelectron spectroscopy. The recording XP spectra for this sample can explain the form of Bi species on the external surface in the bimetallic Pd-Bi catalysts, especially the presence of Bi0 species.
Have you tried to use the membrane to the separation of the catalyst from the reaction mixture. Have you tried to complete the solution of reaction mixture after each injection of HPLC analysis?
Please, can you explain, how did you assume the content of base (NaOH) and the stirring speed (1500 rpm) which were applied to the catalytic tests? Did you show the texture properties of catalysts (the total pore volume, the pore size distribution, the micro- and mesoporous volume).
You applied high ratio n(glucose)/n(Pd) for catalytic test and the fastest conversion in the case of PdBi/Al2O3 was achieved when the content of catalyst was the lowest (you assumed for this catalytic test to apply the molar ration 1 mol of glucose and 600 mol of Pd in catalyst). For the second catalyst, I can not estimate the data, because you don’t show the graph in Figure 10b. Please, can you explain this phenomenon? When you applied higher content of catalyst (e.g. 1 mol of glucose and 3800 mol of Pd in the catalyst), the conversion of glucose in its oxidation was achieved lower at the same time.
Kind regards,
Reviewer
Author Response
Dear Reviewer,
thank you very much for the reviewing of our manuscript titled as Influence of the method of preparation of the Pd-Bi/Al2O3 catalyst on catalytic properties in the reaction of liquid-phase oxidation of glucose into gluconic acid. We have prepared answers to your remarks, suggestions and comments to our work. Please see the attachment.
Sincerely,
Irina A. Kurzina, Prof.
National Research Tomsk State University
Director of the Centre for Research in
Materials and Technologies
Tel: +7-9138821028

Round 2
Reviewer 1 Report
The present paper does not provide a clear evidence for a novelty and a significant advance. Therefore, I do not recommend this as an article in Catalysts.
Author Response
Dear Reviewer,we have made significant amendments to the manuscript in terms of particle formation at each stage of preparation of the catalysts. This will help to choose the method of preparation of the most effective catalyst.
Reviewer 2 Report
Dear Authors,
In my opinion, your corrected manuscript titled: “Influence of the method of preparation of the Pd-Bi/Al2O3 catalyst on catalytic properties in the reaction of liquid-phase oxidation of glucose into gluconic acid” can be published in Catalysts after the second major revision.
Please, you can find below remarks and comments to your work.
In the case of the presented samples, in my opinion, you have to record the XP spectra of the catalysts after their oxidation in the flow of oxygen and the reduction by hydrogen at a selected temperature (at around 100oC, 150oC, 350oC and 500oC). The TPO and TPR studies can show you the possible changes in the chemical composition of catalysts, especially on the external surface.
In my opinion, you have to be careful if you try to the propose of the mechanism of catalytic glucose oxidation based mainly on the products of the reaction without any characterisation of materials after the catalytic tests using, e.g. TEM and/or XPS. This characterisation can confirm also the potential changes in the texture and structure of catalysts.
In my opinion, if you want to compare the catalytic properties of catalysts you have to perform the catalytic test for the same molar ratio, e.g. n(glucose)/n(Pd) = 5000. This can show you, which the method of preparation leads to obtain a more active catalyst.
Kind regards,
Reviewer
Author Response
Dear Reviewer, thanks for your recommendations! We have prepared an answer, please see the attachment

Round 3
Reviewer 1 Report
The revised manuscript is highly improved. I can accept the publication of this paper
Reviewer 2 Report
Dear Authors,
I would like to congratulate you because, in my opinion, your corrected manuscript titled: ''Influence of the method of preparation of the Pd-Bi/Al2O3 catalyst on catalytic properties in the reaction of liquid-phase oxidation of glucose into gluconic aid'' can be published in Catalysts journal. I would like to suggest only the editing correction of manuscript, especially the position of selected figures and tables in the text.
Kind regards,
Reviewer